# Cell-Free Expression of a Therapeutic Protein Serratiopeptidase

**DOI:** 10.3390/molecules28073132

**Published:** 2023-03-31

**Authors:** Yaru Meng, Miaomiao Yang, Wanqiu Liu, Jian Li

**Affiliations:** 1School of Physical Science and Technology, ShanghaiTech University, Shanghai 201210, China; 2Clinical Pathology Center, The First Affiliated Hospital of Anhui Medical University, Hefei 230012, China; 3Anhui Public Health Clinical Center, Hefei 230012, China

**Keywords:** cell-free protein synthesis, therapeutic proteins, serratiopeptidase, enzymatic activity, cell-free synthetic biology

## Abstract

Serratiopeptidase is a clinical therapeutic protein for the treatment of human diseases such as arthritis, bronchitis, and thrombosis. Yet production of this protein in a heterologous host (e.g., *Escherichia coli*) is difficult due to the issue of protein insolubility and the requirement of laborious refolding procedures. Cell-free protein synthesis (CFPS) systems, derived from crude cell extracts, are effective platforms for the expression of recombinant proteins in vitro. Here, we report a new method to produce serratiopeptidase by using an *E. coli*-based CFPS system. After rational selection of cell extracts and construction of expression vectors, soluble expression of serratiopeptidase was achieved and the enzyme activity could be readily tested in the cell-free reaction mixture. By further optimizing the key parameters, optimum conditions for the enzyme activity assay were obtained, including the pH value at 5, reaction temperature at 45 °C, substrate concentration at 10 mg/mL, and supplementing Ca^2+^ ions at 5 mM. Moreover, the CFPS mixture was freeze-dried and the activity of serratiopeptidase could be regenerated by hydration without losing activity. Overall, the CFPS system enabled soluble expression of serratiopeptidase with catalytic activity, providing a new and promising approach for this enzyme production. Our work extends the utility of the cell-free platform to produce therapeutic proteins with clinical applications.

## 1. Introduction

Therapeutic protein-based drugs have been widely used in clinical treatment [1,2,3,4]. Serratiopeptidase, naturally produced by *Serratia marcescens* [5,6], is a proteolytic enzyme with anti-inflammatory, anti-edema, and analgesic effects [7,8,9,10], which has been used to treat arthritis and bronchitis [2,9]. It also has a strong function of dissolving thrombi and is used for the prevention and treatment of thrombosis [11]. In addition, due to its pharmaceutical properties, serratiopeptidase can also be a potentially useful adjuvant for management in the current COVID pandemic [12,13]. Serratiopeptidase is a metalloproteinase with a divalent zinc ion located in the catalytic activity center of the N-terminal domain, and its C-terminal domain can bind to seven calcium ions to promote protein folding [14]. Overproduction of serratiopeptidase using the native producer *S. marcescens* is challenging due to the strain’s pathogenicity, lacking genetic tools to engineer the strain, and the difficulty to cultivate *S. marcescens* [15]. While *E. coli* has been used for heterologous production, serratiopeptidase shows cytotoxicity on the host cell. Moreover, serratiopeptidase can only be expressed as insoluble inclusion bodies and subsequent processes such as protein refolding and purification are required to regenerate active soluble serratiopeptidase [16]. Therefore, developing a new method for directly soluble expression of serratiopeptidase is highly desirable.

Cell-free protein synthesis (CFPS) systems have shown wide applications in biotechnology and synthetic biology and have become a powerful platform for in vitro protein synthesis. Crude extract-based CFPS systems synthesize target proteins by harnessing the intracellular catalysts (e.g., aminoacyl-tRNA synthetases, RNA polymerases, ribosomes, chaperones, etc.) that are necessary for transcription, translation, and protein folding from crude cell lysates of microbial, plant, or mammalian cells. When combined with necessary substrates, which include nucleotides, energy substrates, amino acids, gene templates, cofactors, and salts, these biological catalysts can work as a chemical factory to carry out sustained protein synthesis. The CFPS system has several notable advantages [17,18,19]. First, in CFPS reactions, no living cells are required that can eliminate the influence of cytotoxicity on host cells. Second, gene templates (plasmid or linear DNA templates) can be used directly in CFPS for protein synthesis. Third, the reaction environment is open and thus, can be easily manipulated and optimized. Recently, CFPS has been used for various protein production such as antibodies [20], metalloproteins [21], large proteins [22,23], membrane proteins [21,24], and virus-like particles [25]. In addition, CFPS-based systems have been used for gene circuit construction [26,27], metabolic pathway prototyping [28,29,30], natural product biosynthesis [31,32,33], and medical diagnosis [34,35].

In this work, we aim to express serratiopeptidase by using a well-developed *E. coli*-based CFPS system (Figure 1). First, we expressed serratiopeptidase in the CFPS system and optimized different factors to facilitate its soluble expression. Then, the activity of serratiopeptidase synthesized in CFPS was detected and the conditions that might impact on the activity were explored. Finally, the cell-free reaction mixture containing serratiopeptidase was lyophilized, which then could be readily hydrated showing enzyme activity. Looking forward, we envision that CFPS can be used to rapidly synthesize other high-value therapeutic proteins when cell-based expression systems remain difficult or not amenable.

## 2. Results and Discussion

### 2.1. Cell-Free Expression of Serratiopeptidase

To begin, an *E. coli*-based CFPS system was chosen for the expression of serratiopeptidase because this system has been well-developed and widely used to express various proteins in vitro [36,37,38,39]. Initially, the strain *E. coli* BL21 Star (DE3) was used for cell extract (CE) preparation to conduct CFPS reactions. The plasmid of pJL1-serratio was used as the gene template for protein expression. However, only insoluble serratiopeptidase was obtained in the CE-derived CFPS reaction (Figure 2a). Then, two chaperones-enriched cell extracts were prepared, which are CE2 (CE contains GroES-GroEL-tig) and CE8 (CE contains DnaK-DnaJ-GrpE + GroES-GroEL). They have been shown to increase the solubility of cell-free expressed proteins [33]. Using CE2 and CE8, the expression was still not satisfactory with no soluble fraction in CE2 and completely no protein expression in CE8 (Figure 2a). Next, we sought to fuse the SUMO tag to the N-terminus of serratiopeptidase, trying to enhance the protein solubility [40,41,42]. In particular, the plasmid of pET28a-SUMO-serratio was constructed for the protein expression. Meanwhile, three cell extracts (i.e., CE, CE2, and CE8) were used to express SUMO-fused serratiopeptidase for comparison. By doing this, we clearly observed the soluble expression of SUMO-serratiopeptidase with the correct molecular weight by using the CE8 cell extract (Figure 2b). Therefore, the construct of pET28a-SUMO-serratio and CE8 were used in our following experiments. Taken together, our results indicate that the CFPS system is a promising platform for the expression of “difficult-to-express” proteins such as the therapeutic serratiopeptidase, which can only be expressed in vivo (e.g., *E. coli*) with insoluble inclusion bodies that need laborious refolding and purification steps. By contrast, CFPS systems are open and flexible, and can be easily manipulated and optimized for protein expression.

### 2.2. Activity Assay of Cell-Free Expressed Serratiopeptidase

Having realized soluble expression of serratiopeptidase, we next wanted to test if the cell-free expressed serratiopeptidase exhibits bioactivity as a proteolytic enzyme. For this activity assay, azocasein was selected as a nonspecific protease substrate, which can be hydrolysis by serratiopeptidase to yield TCA-soluble azopeptides with absorbance at around 450 nm [16]. After protein expression, the CFPS reaction mixture was mixed with the substrate (azocasein) solution and incubated at 45 °C for 30 min, followed by adding the TCA solution to terminate the reaction and measuring the absorbance at 450 nm. Meanwhile, three control experiments were performed for comparison. In the first control, no gene template was added to the CFPS reaction and no azocasein was added for the activity assay. In the second control, only the gene template was added without the substrate in the reaction. In the third control, no gene template was added to the CFPS reaction, but the substrate azocasein was added for the activity assay. As can be seen in Figure 3a, the experimental group (+plasmid, +substrate) showed notably higher enzyme activity compared to the three control groups. No activity was detected in the first two controls. Although the third control group also showed clear enzyme activity, it was about 6 times lower than the experimental group. This background enzyme activity in the third control is likely due to the endogenous proteases/peptidases present in *E. coli* cell extracts. As a result, the significant increase of enzyme activity in the experimental group was ascribed to the activity of cell-free expressed serratiopeptidase. Moreover, we further incubated the enzymatic reaction for a total of 7 h and observed that the enzyme could keep catalytic activity for up to 5 h (Figure 3b). Overall, our data suggest that cell-free expressed serratiopeptidase is active and the activity can be directly tested by using crude enzyme without purification, which also streamlines the process for developing serratiopeptidase.

### 2.3. Effect of Physicochemical Factors on Serratiopeptidase Activity

Next, we sought to optimize the physicochemical factors for the activity assay using cell-free expressed serratiopeptidase. The reaction factors of pH value, temperature, substrate concentration, and metal ions that might influence the enzyme activity were investigated. First, the pH value of the enzymatic reaction was tested ranging from 4 to 10. The results indicated that the enzyme favored a slightly acidic condition for the catalysis with the optimum pH values between 5 and 6 (Figure 4a). When the pH value was further increased, a rapid linear decrease of the enzyme activity was observed from pH 7 to pH 10. Reaction temperature is another key factor requiring optimization because it often affects enzyme’s activity. Thus, a broad range of temperature from 30 to 60 °C were compared. After incubation of the enzymatic reaction at each temperature, we found the best temperature was 45 °C, in which the activity was 4.4 times higher than the reaction at 30 °C (Figure 4b). In general, high temperatures (40–55 °C) were better than low temperatures (30 and 37 °C) for serratiopeptidase activity. Next, the substrate (azocasein) concentration in the activity assay reaction was evaluated. The results showed that the highest enzyme activity was observed when 10 mg/mL of azocasein was used in the reaction (Figure 4c). Further increase of the azocasein concentration (>15 mg/mL) led to a low, stable level of enzyme activity. Finally, we were curious if metal ions have impacts on the enzyme activity because serratiopeptidase is a metalloenzyme requiring both Zn^2+^ and Ca^2+^ [14,43]. To this end, additional metal ions (i.e., Zn^2+^, Ca^2+^, Mg^2+^, Cu^2+^, Mn^2+^, Na^+^, and K^+^) were added to the activity assay reactions. A control reaction was also performed without adding additional metal ions. We found that the enzyme activity could be increased by 2.3 times by supplementation of Ca^2+^ in the assay (Figure 4d). This is likely due to the fact that calcium ions can keep the enzyme more stable as seven Ca^2+^ ions are required to facilitate the folding of serratiopeptidase [14,43]. Yet, Zn^2+^ and other tested metal ions (except Cu^2+^ which is notably deleterious to the assay) did not obviously increase the enzyme activity. While one Zn^2+^ ion is needed to be in the catalytic center of serratiopeptidase, Zn^2+^ ions in the CFPS reaction mixture might be already sufficient to be reconstituted in the catalytic center. Thus, adding more Zn^2+^ ions in the assay reaction did not further help increase the activity of serratiopeptidase. The above optimized parameters will be useful for rapidly monitoring the enzyme activity when using the CFPS system to express serratiopeptidase. This is important because (i) the expression level of serratiopeptidase can be easily evaluated according to the activity assay, and (ii) the production process can be monitored on a timely basis to achieve the highest productivity with soluble and active serratiopeptidase.

### 2.4. Lyophilization of Cell-Free Expressed Serratiopeptidase

Lyophilization (also known as freeze-drying) is a method to preserve labile biological materials in a dehydrated form, which has been widely used for the long-term storage of high-value biomolecules such as therapeutic proteins [44]. We, therefore, wanted to test the effect of lyophilization on cell-free expressed serratiopeptidase. To do this, CFPS reactions were performed to express serratiopeptidase. Then, the reactions were mixed and divided into two groups with different total volumes (90 μL and 180 μL) for the following lyophilization process. After lyophilization, the residual mixture was hydrated by adding either 90 μL of acetate buffer (pH 5.0) or 90 μL of nuclease-free water (NFW). Then, the enzyme activity was measured in each group. In addition, a fresh CFPS reaction without lyophilization was used as a control for comparison. The data suggested that the lyophilization process had no detrimental impact on the enzyme activity and both hydrating solutions (buffer and NFW) could regenerate the activity to almost the same level (Figure 5). The group with a total volume of 180 μL reaction mixture, which was freeze-dried and then hydrated with a half volume of buffer or water (namely, in 90 μL), showed a high activity due to the expressed enzymes being concentrated in the final sample for activity assay. Again, their activities were also comparable after hydration with buffer and water, respectively. As a result, lyophilization is a potential method to store cell-free expressed high-value products like the example of the therapeutic serratiopeptidase showcased in this work.

## 3. Materials and Methods

### 3.1. Strains, Media, and Plasmids

*E. coli* DH5α is used for molecular cloning and plasmid propagation. *E. coli* BL21 Star (DE3) is used in the preparation of extracts. LB medium contains 10 g/L tryptone, 5 g/L yeast extract, and 10 g/L NaCl. 2× YTPG medium contains 10 g/L yeast extract, 16 g/L tryptone, 5 g/L NaCl, 7 g/L K_2_HPO_4_, 3 g/L KH_2_PO_4_, and 18 g/L glucose.

The gene of serratiopeptidase (GenBank: KR014114.1) was codon-optimized for *E. coli* and cloned into pET28a by GENEWIZ (Suzhou, China), yielding pET28a-serratio. Specifically, the serratiopeptidase gene was inserted between two restriction sites of *Nde*I and *Sal*I located in the multiple cloning sites of the plasmid pET28a. To construct pJL1-serratio, the serratiopeptidase gene, flanking with *Nde*I and *Sal*I sites was initially amplified by polymerase chain reaction (PCR) from the template plasmid pET28a-serratio. Then, the PCR-amplified gene fragment and the vector backbone (pJL1, Addgene #69496) were digested with the same restriction enzymes (*Nde*I and *Sal*I), respectively. Afterward, the digested fragment and backbone were ligated, generating pJL1-serratio. If needed, a SUMO tag was fused to the N-terminus of serratiopeptidase for enhanced expression [40,41,42]. The commercial plasmid pET28a-SUMO (Biofeng, Shanghai, China) was used as a backbone for inserting the serratiopeptidase gene between *Sac*I and *Hind*III. In brief, the serratiopeptidase gene, flanking with *Sac*I and *Hind*III, was PCR-amplified from pET28a-serratio. After restriction enzyme digestion, the gene fragment and the vector backbone were ligated, giving rise to the plasmid pET28a-SUMO-serratio. The primers used for PCR amplification were listed in Table 1. The correctness of all constructs was validated by DNA sequencing (GENEWIZ, Suzhou, China).

### 3.2. Cell Extract Preparation

Cell cultivation, harvest, and lysis were prepared according to our previous report [33,45]. Briefly, *E. coli* BL21 Star (DE3) cells were grown in 1 L 2x YTPG medium in 2.5 L baffled Ultra Yield™ flasks (Thomson Instrument Company, Oceanside, CA, USA). After inoculation with 20 mL overnight preculture (initial OD_600_ of 0.05), the cells were incubated in the shaker at 220 rpm and 34 °C. When the OD_600_ reached 0.6–0.8, cells were induced with 1 mM isopropyl-β-D-thiogalactopyranoside (IPTG) to express T7 RNA polymerase. To prepare cell extracts consisting of chaperones, all cultures were induced with 0.8 mM IPTG. Note that the final concentration of IPTG was reduced to 0.8 mM according to the manufacturer’s manual (Chaperone Plasmid Set, Code No. 3340, TaKaRa, Kusatsu, Japan). In addition, 1 mg/mL L-arabinose and 1 ng/mL tetracycline were added to induce pG-KJE8 for the expression of DnaK-DnaJ-GrpE and GroES-GroEL; and 1 ng/mL tetracycline was added to induce pG-Tf2 for the expression of GroES-GroEL-tig. Then, cells were harvested at an OD_600_ of 3.0 by centrifugation at 5000× *g* and 4 °C for 15 min. Afterward, cell pellets were washed three times with cold S30 Buffer (10 mM Tris-acetate, 14 mM magnesium acetate, 60 mM potassium acetate, and 2 mM dithiothreitol (DTT)). After that, the pellet was resuspended in S30 Buffer (1 mL per gram of wet cell biomass) and lysed by sonication (10 s on/off, 50% of amplitude). The lysate was then centrifuged twice at 12,000× *g* and 4 °C for 10 min. The resulting supernatant was flash frozen in liquid nitrogen and stored at −80 °C until use.

### 3.3. Cell-Free Protein Synthesis (CFPS) and Western-Blot Analysis

A standard CFPS reaction was conducted in a 1.5 mL microcentrifuge tube. Each reaction (15 μL) contains the following components: 12 mM magnesium glutamate, 10 mM ammonium glutamate, 130 mM potassium glutamate, 1.2 mM ATP, 0.85 mM each of GTP, UTP, and CTP, 34 μg/mL folinic acid, 170 μg/mL of *E. coli* tRNA mixture, 2 mM each of 20 standard amino acids, 0.33 mM nicotinamide adenine dinucleotide (NAD), 0.27 mM coenzyme A (CoA), 1.5 mM spermidine, 1 mM putrescine, 4 mM sodium oxalate, 33 mM phosphoenolpyruvate (PEP), 13.3 μg/mL plasmid, and 27% (*v*/*v*) of cell extract. CFPS reactions were incubated at 30 °C for 2.5 h unless otherwise noted.

After CFPS reactions, all cell-free expressed proteins were analyzed by Western-blot. To do this, the reaction mixture was first centrifuged at 4 °C and 12,000× *g* for 10 min, and the supernatant was collected as soluble fractions. Then, 10 μL of total protein (without centrifugation) and 10 μL of soluble protein (supernatant) samples were mixed with 10 μL of 2× loading buffer, respectively, and heated at 98 °C for 10 min. Afterward, each heated sample was loaded to SDS-PAGE gel for protein separation, followed by wet transferring to PVDF membrane (Bio-Rad, Hercules, CA, USA) with 1× transfer buffer (25 mM Tris-HCl, 192 mM glycine, and 20% methanol, pH 8.3). Then, the PVDF membrane was blocked with Protein-Free Rapid Blocking Buffer (EpiZyme, Shanghai, China) for 30 min at room temperature. After washing thrice with 1x TBST buffer (10 mM Tris-HCl, 150 mM NaCl, and 0.1% Tween 20, pH 7.5) for each 5 min, 1:10,000 TBST buffer-diluted His-Tag Mouse Monoclonal Antibody (Proteintech, Rosemont, IL, USA) solution was added to the membrane and incubated for 1 h at room temperature. After washing again with 1× TBST buffer thrice for each 5 min, 1:10,000 TBST buffer-diluted HRP-Goat Anti-Mouse IgG (H+L) Antibody (Proteintech) solution was added to the membrane and incubated for another 1 h at room temperature. After the last washing with TBST thrice for each 5 min, the PVDF membrane was visualized using Omni ECL reagent (EpiZyme, Shanghai, China) under UVP ChemStudio (Analytik Jena, Jena, Germany).

### 3.4. Activity Assay of Serratiopeptidase

Azocasein (Sigma, St. Louis, MO, USA) was used as a substrate for determining the activity of serratiopeptidase. Briefly, 90 μL of the CFPS reaction mixture (containing cell-free expressed serratiopeptidase) was mixed with 210 μL of the azocasein solution (10 mg/mL dissolved in nuclease-free water), followed by incubation at 45 °C for 30 min. Then, 113 μL of 20% trichloroacetic acid (TCA) was added to stop the reaction, which was then centrifuged at 10,000× *g* for 5 min. Afterward, 100 μL of the supernatant was transferred to a 96-well plate and the NaOH solution (1 N) was added to the supernatant at a 1:1 (*v*/*v*) ratio in each well. Then, the absorbance at 450 nm (OD_450_) was measured using a microplate reader. One enzyme unit (U) is defined as an increase of 0.01 absorption unit after 30 min of incubation at 45 °C.

### 3.5. Effect of Reaction Time on Serratiopeptidase Activity

After the reaction, the CFPS mixture was mixed with azocasein solution (10 mg/mL) and incubated at 45 °C for 0.25, 0.5, 1, 2, 3, 5, and 7 h, respectively. Then, the absorbance at OD_450_ was measured accordingly. A mixture of CFPS reaction without plasmid was used as a negative control.

### 3.6. Effect of Physicochemical Factors on Serratiopeptidase Activity

Four physicochemical factors of the enzymatic reaction were investigated. First, the reaction pH values were evaluated at 4, 5, 6, 7, 8, 9, and 10, respectively. The enzymatic reaction was incubated at 45 °C for 30 min and the absorbance at OD_450_ was measured. Second, the reaction temperature was tested at 30, 37, 40, 45, 55, and 60 °C, respectively. Each reaction was incubated for 30 min and the absorbance at OD_450_ was measured accordingly. Third, the substrate (azocasein) concentrations were set at 0.1, 1, 5, 10, 15, and 30 mg/mL, respectively. Each reaction was incubated at 45 °C for 30 min and the absorbance at OD_450_ was measured accordingly. Finally, the effect of different metal ions on the serratiopeptidase activity was investigated. The metal ions of Zn^2+^, Ca^2+^, Mg^2+^, Cu^2+^, Mn^2+^, Na^+^, and K^+^ were added to the enzymatic reactions. The final concentration of each ion in the reaction was 5 mM. After adding each ion to the CFPS mixture and standing at room temperature for 1 h, the substrate (azocasein) solution was mixed with CFPS for the activity assay. Each reaction was incubated at 45 °C for 30 min and the absorbance at OD_450_ was measured accordingly.

### 3.7. Detection of Serratiopeptidase Activity after Lyophilization

CFPS reactions (90 μL and 180 μL) containing expressed serratiopeptidase were freeze-dried overnight in 1.5 mL microcentrifuge tubes, respectively. Then, the lyophilized mixtures were hydrated with 90 μL reaction buffer (pH 5) or 90 μL nuclease-free water. The resulting solution was then mixed with the substrate (azocasein) solution and incubated at 45 °C for 30 min for enzyme activity assay. The absorbance at OD_450_ was measured accordingly.

### 3.8. Statistical Analysis

Values show means with error bars representing standard deviations (s.d.) of at least three independent experiments. Student’s *t*-tests were used for statistical analysis, and *p* < 0.05 indicated statistical significance (* *p* < 0.05, ** *p* < 0.01, and *** *p* < 0.001).

## 4. Conclusions

In this study, an *E. coli*-based CFPS system was used to express serratiopeptidase—a therapeutic protein. Heterologous expression of serratiopeptidase in a surrogate host such as *E. coli* made it difficult to obtain soluble proteins [16]. By using the cell-free system, soluble expression of serratiopeptidase was achieved after the rational selection of cell extracts and construction of expression vectors. The enzyme activity could also be detected with the crude enzyme mixture without purification, which makes the monitoring and optimization processes easier. After optimization, the key parameters for the enzyme activity assay were obtained, including the optimum reaction conditions of pH value at 5, temperature at 45 °C, substrate concentration at 10 mg/mL, and supplementing Ca^2+^ ions at 5 mM. Finally, the CFPS reaction mixture was freeze-dried and the enzyme activity of serratiopeptidase could be easily regenerated by hydration without losing activity. Taken together, the CFPS system enables soluble expression of serratiopeptidase with catalytic activity, providing a new and promising approach for this enzyme production. As a result, our work demonstrates that CFPS is an efficient platform for the expression of “difficult-to-express” therapeutic proteins. On the one hand, the expression conditions (e.g., gene constructs, molecular chaperons, reaction temperature, etc.) and enzyme activity can be rapidly optimized by using the CFPS system. On the other hand, purification of the proteins in large amounts might also be possible in the near future as the *E. coli* CFPS system has been shown to scale up linearly from 250 μL to 100 L reaction volumes [46]. Looking forward, we envision that the *E. coli* CFPS system together with various other CFPS platforms [47,48,49,50,51,52,53,54,55,56,57,58] will make significant contributions in the future for making not only proteins of pharmaceutical and industrial importance, but also valuable chemicals and materials for compelling applications.

## Figures and Tables

**Figure 1 molecules-28-03132-f001:**
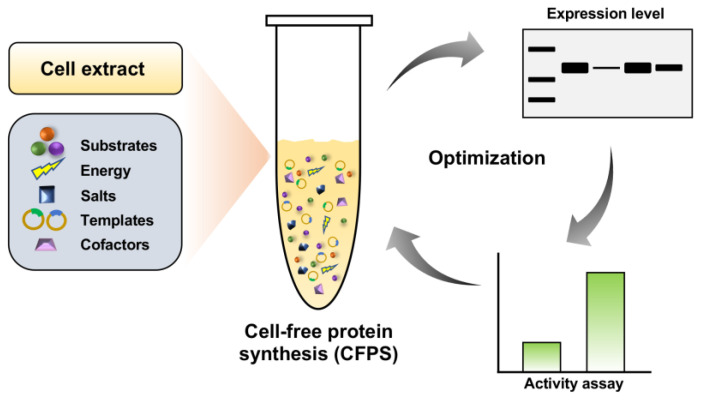
Schematic diagram of cell-free protein synthesis (CFPS) and optimization for soluble and active expression of serratiopeptidase.

**Figure 2 molecules-28-03132-f002:**
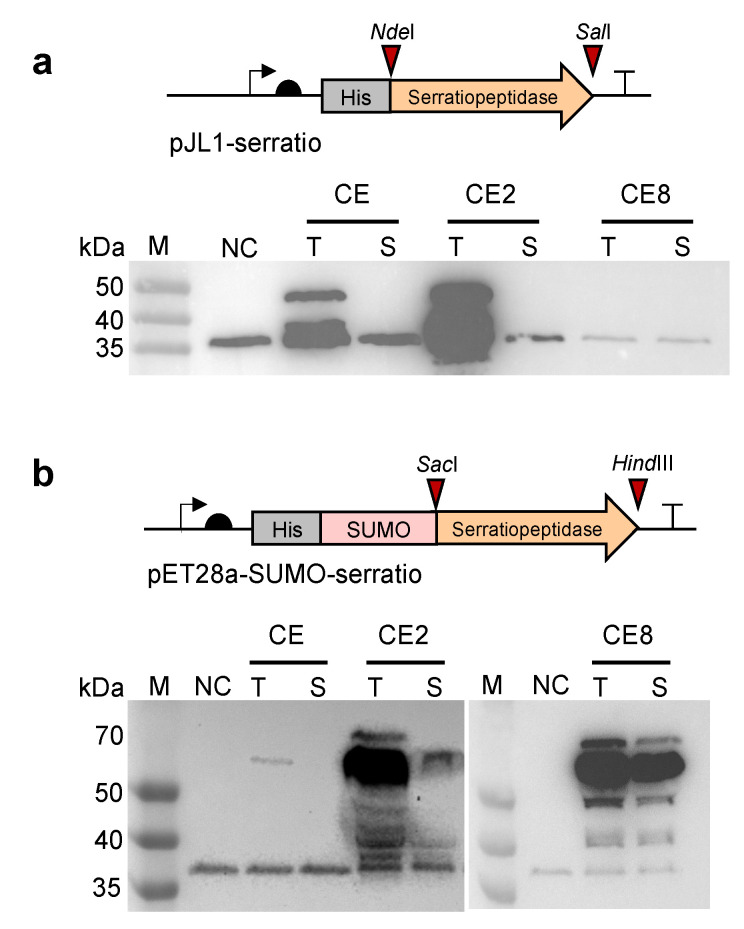
Optimization of the CFPS system for soluble expression of serratiopeptidase. (**a**) Western-blot analysis of cell-free expressed serratiopeptidase (50 kDa) using different cell extracts. The plasmid of pJL1-serratio with an N-terminal 6x His tag was used for CFPS reactions. (**b**) Western-blot analysis of cell-free expressed SUMO-serratiopeptidase (61 kDa) using different cell extracts. The plasmid of pET28a-SUMO-serratio with an N-terminal 6x His tag was used for CFPS reactions. In (**a**,**b**), CE: cell extracts prepared from *E. coli* BL21 Star (DE3); CE2: CE enriched with the chaperones of GroES-GroEL-tig; CE8: CE enriched with the chaperones of DnaK-DnaJ-GrpE and GroES-GroEL; NC: negative control without plasmid in the CFPS reaction; T: total protein; S: soluble protein; M: protein molecular weight marker. The protein band between 35 and 40 kDa was an unknown protein derived from the cell lysate.

**Figure 3 molecules-28-03132-f003:**
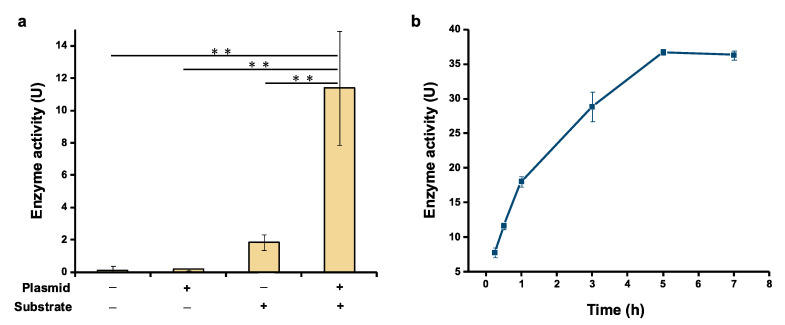
Detection of the serratiopeptidase activity. (**a**) Confirming the activity of cell-free expressed serratiopeptidase. (**b**) Time course of the enzyme activity measured within 7 h incubation. One enzyme unit (U) is defined as an increase of 0.01 absorption (at 450 nm) unit after 30 min of enzymatic reaction at 45 °C. Values show means with error bars representing standard deviations (s.d.) of at least three independent experiments. Student’s *t*-tests were used for statistical analysis, and *p* < 0.05 indicated statistical significance (** *p* < 0.01).

**Figure 4 molecules-28-03132-f004:**
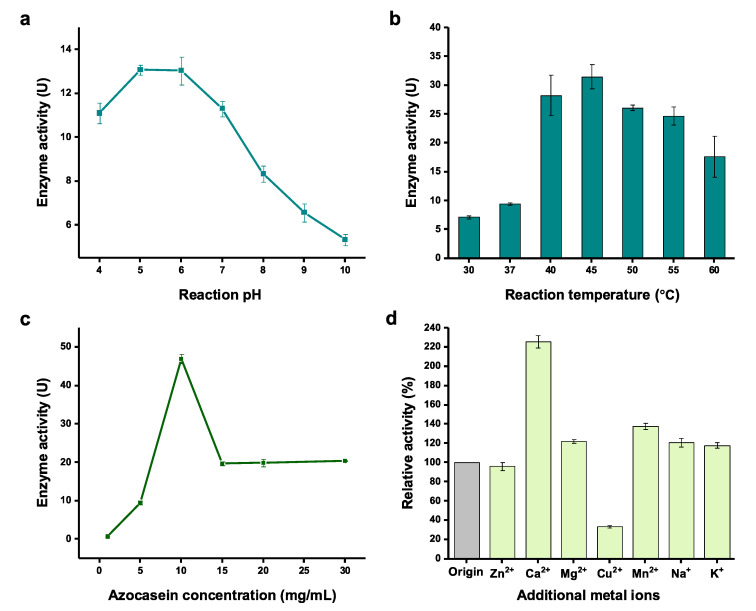
Optimization of physicochemical factors for the assay of serratiopeptidase activity. Effects of (**a**) reaction pH, (**b**) reaction temperature, (**c**) azocasein concentration, and (**d**) metal ions on the serratiopeptidase activity. Note that each value was calculated by subtracting the background activity. Values show means with error bars representing standard deviations (s.d.) of at least three independent experiments.

**Figure 5 molecules-28-03132-f005:**
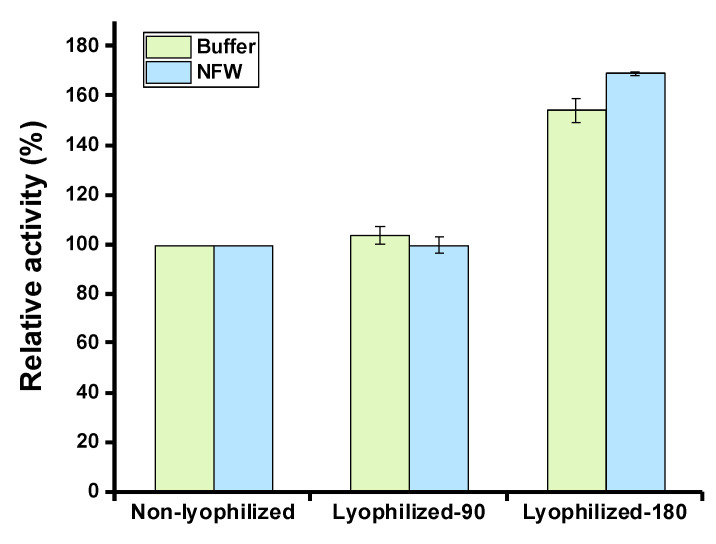
Effects of lyophilization on the serratiopeptidase activity. Lyophilized-90: 90 μL of cell-free reaction mixture was freeze-dried in one tube; Lyophilized-180: 180 μL of cell-free reaction mixture was freeze-dried in one tube; Non-lyophilized: cell-free reaction mixture was not freeze-dried; Buffer: acetate buffer (pH 5.0); NFW: nuclease-free water. Values show means with error bars representing standard deviations (s.d.) of at least three independent experiments.

**Table 1 molecules-28-03132-t001:** Primers used for PCR amplification of the serratiopeptidase gene.

Construction of Plasmid	Primer (5′→3′)	Restriction Site ^a^
pJL1-serratio		
Forward	CATCATCATCATCACCATATGGCGGCGACCACCGGCTATGATG	*Nde*I
Reverse	TTTGTTAGCAGCCGGTCGACTTACACAATAAAATCGGTCGCCAC	*Sal*I
pET28a-SUMO-serratio		
Forward	GATTGGTGGTACCGAGCTCATGGCGGCGACCACCGGCTATG	*Sac*I
Reverse	CGAGTGCGGCCGCAAGCTTTTACACAATAAAATCGGTCG	*Hind*III

^a^ Restriction sites are underlined in the primer sequences.

## Data Availability

Not applicable.

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
