# Peer review of "Cell-Free Expression of a Therapeutic Protein Serratiopeptidase"

_molecules, 2023, doi:10.3390/molecules28073132_

Round 1

Reviewer 1 Report

In general, the manuscript needs important corrections:
1. The reaction must be in third person.

2. The first conclusion (Page 7, Line 256-257) could not obtain with the experimental design used.

3. The discussion is poor, and it needs to be enriched.

Reviewer 2 Report

The authors of the work describe with a biotechnological approach the development of the best conditions for the expression and functionality of the protein serratiopeptidase which up to now has encountered various limitations and production difficulties. Specifically, the authors of the work describe the in vitro production of the protein using the well-known cell free protein system based on the S30 fraction of the E.coli strain BL21 Star (DE3). The work is described in a linear way, step by step, showing the experimental approach adopted both to solubilize the serratiopeptidase and the best physical-chemical conditions of the system to improve the proteolytic activity of the enzyme. Finally, the authors describe the freeze-drying conditions of the expressed protein without it losing enzymatic activity with respect to the non-lyophilized original, suggesting this practice as a potential method of product preservation for its future therapeutic application. Despite the ease of reading, the work show some shortcomings in the methodological description of some key steps in the results obtained. For these reasons, the scientific community would not have the opportunity to put itself in the experimental conditions described by the authors in order to reproduce the results of the work. Therefore, it is strongly recommended that the authors clarify the points listed below in order to make the work shareable with the rest of the scientific community.

Specifically:

1)    The authors describe the production of the "pET28a-serratio" construct by limiting the description with the word "by GENEWIZ". Both the cloning sites used in the MCS (multiple cloning sites) of the plasmid and sequences of the oligonucleotide used to amplify and clone the serratiopeptidase gene are not described. Add a detailed description of the cloning procedure, please. The same goes also for the "pJL1-serratio" construct.

2)    The authors mention the production of the "pET28a-SUMO-serratio" construct via the addition of a SUMO tag in the N-terminal region of the protein without citing the literary sources that rationalize its use (i.e. "Panavas T, Sanders C, Butt TR. SUMO fusion technology for enhanced protein production in prokaryotic and eukaryotic expression systems. Methods Mol Biol. 2009; 497:303-17. doi: 10.1007/978-1-59745-566-4_20. PMID: 19107426".) This quote is an example of a work summarizing the benefits of using the sumo-tag to solubilize proteins but feel free to use any other similar quote in the description in place of my example. Also for this construct, which restriction sites were used to add the SUMO-tag? What is the sequence of oligonucleotides used to add the tag? The authors have previously published papers describing the in vitro expression of other molecules using a CFPS approach similar to the one described here. In this work they also mentioned some of their past publications (i.e. Tian X, Liu WQ, Xu H, Ji X, Liu Y, Li J. Cell-free expression of NO synthase and P450 enzyme for the biosynthesis of an unnatural amino acid L- 4-nitrotryptophan. Synth Syst Biotechnol. 2022 Mar 23;7(2):775-783. doi: 10.1016/j.synbio.2022.03.006. PMID: 35387232; PMCID: PMC8956912.").  In "Materials and Methods" of their above citated work, they report a table with the list and sequence of used oligonucleotides. Why not do the same now?

3)    Figure 2 (a, b) shows the scheme of the constructs used. Replace the construct name “pJL1” with “pJL1-serratio” and the name “pET28a” with “pET28a-SUMO-serratio”, please. The name of the plasmid alone can be confusing. The scheme shows a vector with an insert and therefore it is clearer to report the full name of the construct next to the scheme. It would also be useful to share the restriction sites used for cloning in the construct schema.

4)The experiment represented in fig. 3a lacks the plasmid-only control in the CFPS without the substrate. Add the data of this reaction to the graphic, please.

5) All graphs regarding the quantization of measures such as enzyme activity and relative activity (Fig. 3(a) and (b), Fig. 4(a–d), Fig. 5) show an error bar. However, the description of the statistical analysis is missing both in the legend of the figures and in the Materials and Methods section. Furthermore, in Fig. 3a, the enzyme activity data derived from the plasmid + substrate incubation show one standard deviation that requires a statistical test of significance based on ANOVA or similar. Take action in this regard, please.

Finally, a comment that does not prejudice the previous opinion, but is a consideration on the production process. Mainly a curiosity: the described system is limited to define the production and the analysis of the activity of the serratiopeptidase in the CFPS. However, for commercialization of the protein, I don't think it can be given to patients with all the E.coli extract. How come you didn't even think of analyzing its purification? Considering you express it with a SUMO tag, its purification shouldn't pose any major problems, I guess!

Reviewer 3 Report

The authors in the manuscript report use of Cell-Free Protein Synthesis (CFPS) of a therapeutic protein Serratiopeptidase. The authors demonstrate several optimization strategies used to achieve protein expression and activity. Overall the manuscript is well presented with methodical experiments and controls. 

Minor points:

1. Abstract -Line 21: Please rephrase the term "active expression". It's not clear if the authors are stating successful expression or functionally active protein.

2. Results and Discussion- Line 75-76: Please elaborate how the two chaperones enriched cell extracts (CE2 & CE8) were prepared in the methods section.

3. Please report average protein concentration of the target protein obtained from the CFPS.

4. Line 118-121: Is there an inhibitor (as a control study) that can demonstrate decreased activity of serratiopeptidase.

5. Figure 3: Please expand on how the enzyme activity units were calculated. 

6. Figure 4: It's not clear if control (+substrate, -plasmid) was used in this study to demonstrate activity changes were specific to serratiopeptidase. This should be addressed as the substrate used is not specific for the target protein. 

7. The reviewer is curious to know what the authors stance is on this system for difficult to express therapeutic proteins: Can this be scaled-up for protein purification or is this a platform to screen activity and expression conditions (e.g. gene constructs, chaperons, temperature etc)? It would be a nice addition in the conclusion section of the manuscript.

Round 2

Reviewer 1 Report

The authors have addressed all the observations made. So the manuscript can be published.

Author Response

Thank you for supporting publication.

Reviewer 2 Report

The authors of the article implemented the required points.

Specifically:

1) They have added a detailed description of the cloning procedure for all constructs used, both in the text of the Results section and in Materials and Methods.

2) Among the references, they mentioned the articles summarizing the benefits of using the sumo-tag to solubilize proteins (cit. 40-42).

3) In "Materials and Methods", they added a table with the list and sequence of used oligonucleotides.

4) In Figure 2 (a, b), they replaced the construct name “pJL1” with “pJL1-serratio” and the name “pET28a” with “pET28a-SUMO-serratio”. Moreover, they added the restriction sites used for cloning in the construct schema of the mentioned figure.

5) The control consisting of plasmid only, with no substrate, was added to Figure 3a.

6) The statistical analysis has been reported in the legend of the figures (whenever necessary) and in the description of Materials and Methods.

Just two more points to correct:

1) Replace the sentence of Lane 117 "There was no activity was detected in the first two controls" with the following one "No activity was detected in the first two controls", please.

2) In table 1 (between lanes 224 and 225), replace the lowercase letters of the primer sequences with the uppercase letters, please.
